# Evidence-based policy-making in sports funding using a data-driven optimization approach

**Jan Hurt**[1]*, **Liuhuaying Yang**[1], **Johannes Sorger**[1], **Thomas J. Lampoltshammer**[2], **Nike Pulda**[2], **Ursula Rosenbichler**[4], **Stefan Thurner**[1,3,5], **Peter Klimek**[1,3]

**1** Complexity Science Hub, Vienna, Austria, **2** University for Continuing Education, Krems, Austria, **3** CeMSIIS, Medical University of Vienna, Vienna, Austria, **4** Federal Ministry for Arts, Culture, the Civil Service and Sport, Vienna, Austria, **5** Santa Fe Institute, Santa Fe, New Mexico, United States of America

\* hurt@csh.ac.at

**Data Availability Statement:** Data and code is available at: https://drive.google.com/file/d/1ePz5j2mKpBaWUa1a-8dDqXC-Ct-nrhHY/view?usp=share_link.

## Abstract

Regular physical activity is essential for the healthy development of children, and sports clubs are one of the main drivers of regular exercise. Previous studies have demonstrated that public subsidies can increase participation rates in sports clubs. The effectiveness of funding in increasing participation rates depends on multiple factors, such as geographic location, the size of the sports club, and the socio-economic conditions of the population. Here, we show how an optimal allocation of government funds to sports facilitators (e.g., sports clubs) can be achieved using a data-driven simulation model that maximizes children's access to sports facilities. We compile a dataset for all 1,854 football clubs in Austria, including estimates for their budgets, geolocations, tallies, and the age profiles of their members. We find a characteristic sublinear relationship between the number of active club members and the budget, which depends on the socio-economic conditions of the club's municipality. In the model, where we assume this relationship to be causal, we evaluate different funding strategies. We show that an optimization strategy, where funds are distributed based on regional socio-economic characteristics and club budgets, outperforms a naive approach by up to 117% in attracting children to sports clubs with 5 million euros of additional funding. Our results suggest that the impact of public funding strategies can be substantially increased by tailoring them to regional socio-economic characteristics in an evidence-based and individualized way.

## Introduction

Regular physical activity is crucial for the healthy development of children [1–6], and voluntary sports clubs play a fundamental role in ensuring consistent participation. In Austria, for example, 40% of children exercise multiple times a week in sports facilities organized by clubs [7]. In Germany, most of the population aged 4-17 years regularly exercises in sports clubs [8]. Many countries subsidize voluntary sports clubs to increase physical activity in

**Funding:** The project was funded by the European Union's Structural Reform Support Programme (SRSP) under Grant Agreement No. GA2020/023 (received by S.T.; https://commission.europa.eu/about-european-commission/departments-and-executive-agencies/structural-reform-support/structural-reform-support-programme-srsp_en), and the Austrian Research Promotion Agency (FFG) under project number P 882184 (received by S.T.; https://www.ffg.at/en). The funders had no role in study design, data collection and analysis, decision to publish, or preparation of the manuscript.

**Competing interests:** The authors have declared that no competing interests exist.

child and adolescent population [9], in line with the WHO recommendations [10]. However, understanding the most effective ways to allocate these subsidies to maximize youth participation remains a critical area of study. This research seeks to explore how different funding strategies can optimize participation in sports, ultimately enhancing the impact of public subsidies.

Studies have demonstrated that public subsidies can significantly increase participation rates in sports clubs, with the availability of sports facilities being a key driver of this effect [9, 11, 12]. However, the question of how to distribute these funds optimally to maximize their impact remains open. The effectiveness of subsidies may vary depending on socio-economic factors, making it crucial to consider these indicators when designing funding strategies. For instance, research in the Netherlands has shown that sports participation rates among 6- to 17-year-olds saw little change for high-income households but increased by 5% for average-income households and by nearly 15% for low-income households when public subsidies were provided [13]. This suggests that targeted funding could play a significant role in addressing disparities in sports participation, particularly among underprivileged groups.

In this work, we estimate the functional relationship between regional sports funding and the participation rates of children in Austrian football clubs. Assuming this relationship to be causal, it becomes possible to derive an optimal funding scheme that drastically outperforms standard funding strategies, such as distributing funds equally across all clubs or proportionally to club size. We illustrate the model in Fig 1(A) and 1(B). Funders distribute subsidies across clubs, and the additional funding allows clubs to attract additional children from the surrounding municipalities.

To this end, we have construct a comprehensive dataset that contains nearly all 1,854 football clubs with a total of approximately 111,000 children. It includes the number of members aged below 18 years, the geocoded location of the pitch, and the estimated budget, which depends on the competition level—i.e., the tier within the domestic hierarchy in which the club's primary adult team competes. Employing a simple gravity model (see Materials and methods for details), we estimate the number of football-playing children across Austria's 8,820 counting districts (CDs, the smallest administrative unit) A map showing the fraction of children playing football in each CD, as presented in panel (C) of Fig 1, visualizes this data. This data allows us to quantify the functional dependence between active club members and budget, as well as the socio-economic factors that influence this relationship. Assuming this relationship to be causal, we extrapolate the *marginal activity rate* as the number of additional children a club can attract with an additional unit of subsidies. The goal is to find the optimal funding strategy that allocates funds to maximize overall activity. In the optimal scheme, the club with the highest marginal activity rate receives funding (in small units) as long as it maintains the highest rate. As more funds are allocated, the marginal activity rate will eventually decrease, and when another club achieves the highest marginal rate, it will receive the funding units. In this manner, funds will always be redirected to the club with the highest return. This procedure is repeated until all available funds are allocated. This optimized funding strategy is then compared to three alternative strategies based on absolute activity. The first allocates funding equally (in absolute terms) among all clubs; in the second, clubs receive funding in proportion to their current budgets; the third is an optimization that neglects socio-economic effects. We present the results, showing how the different strategies vary in their outcomes. In addition, we provide an interactive dashboard https://vis.csh.ac.at/sports-viz/ and an online game https://vis.csh.ac.at/sports-funding-game/ [14].

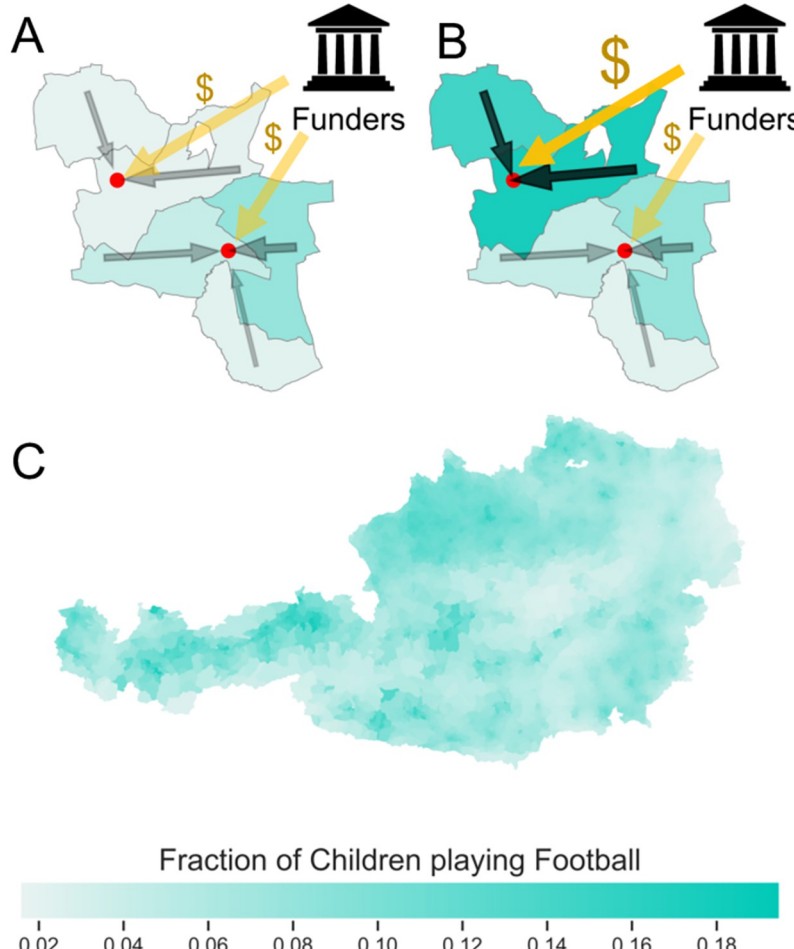

**Fig 1.** (A) A schematic description of the model. Funding agencies distribute funds (yellow arrows marked "$") to regional sports clubs (red dots). These clubs attract children from the surrounding area (black arrows) depending on the amount of funding. We work at the scale of the smallest administrative units, the *counting districts* (CD). The number of children attracted depends on the number of children living in the CD and the distance between the CD and the club. (B) Increasing the funding of a specific club increases its attractiveness and thus the number of children playing football in the surrounding CDs. (C) A map of Austria showing the fraction of children playing football in every CD. There is an urban-rural differential, with more children playing football in urban areas, except for Vienna (the largest city of Austria, which is located in the northeastern area with a low indicator value) and its surroundings.

## Materials and methods

### Population data

Data on the Austrian population are available on three different levels, from the coarsest to the most granular: across 116 districts, 2,115 municipalities, and 8,820 counting districts (CDs). Demographic information for each CD, such as population size, sex and age distribution, and socio-economic indicators for each district, is available at the Austrian National Statistical Office [15, 16]. The latter includes the percentage of people with non-Austrian citizenship, the fraction of people aged 15 and older with secondary or tertiary education, the fraction of those aged 15 to 64 who are employed, the number of commuters, as well as the number of corporations, number of families, available jobs per capita, and household size; this data is available at the municipal level. Political boundaries for CDs are available from open government data

[17]. The area of each CD was calculated from polygon shapes, which were then used to estimate population density. The fraction of children (aged 5-19 years) in relation to the total population was calculated at the district level. To estimate the number of children in each CD, it is assumed that this fraction is constant for all CDs within a given district.

## Clubs

There are 160 football competitions (leagues) in Austria, which are classified into ten tiers. Information on all clubs and their homepages in each competition was collected from the website of the Austrian Football Association (OEFB) [18]. "An automated web scraping algorithm applied to these sites resulted in 1,854 links to club homepages, from which it was possible to retrieve the GPS coordinates of the football pitches and the number of players. Clubs usually have multiple teams, including youth teams, with names that typically signal the age range of the players.("U-XX" where "XX" is the upper bound of the age of the players in this team. For example, a "U-12" team could include players from the ages 10-12 years old.) Some clubs form associations with other clubs to create teams for specific age groups, which can result in individual players being listed multiple times—once for each club in the association. These players are assigned to one club in the association with a probability proportional to club size.

The data collection and cleaning process resulted in a database containing the number of players grouped by age, team, and club. In total, the clubs reported a total of 225,907 active players. Since we are focusing on recreational, non-professional sports clubs, only teams that play in the 3rd tier or lower (higher league number) have been considered for the analysis. Budget information for clubs was stratified by competition level and obtained from a representative survey conducted by SportsEconAustria (SpEA), the Institute for Sports Economy, covering all 1,854 clubs.

## Estimating the number of active children in every CD

The so-called *gravity model* was found to be useful in describing the influence of geographic distance in a number of applications [19], including sports club participation rates [20]. It is assumed that the probability of a child playing football in a specific club, $P(club \rightarrow cd)$, depends on the inverse square distance between the residential CD and the club's location. To avoid problems at a distance of zero, we added 5 km to all distances, $f(r) = \{1/(r + 5\text{km})^2$ *if* $r < 50\text{km}; 0$ else$\}$:

$$P(\text{club} \rightarrow \text{CD}) = \frac{f(d(\text{CD}, \text{club}))}{\sum_{\text{CD}} f(d(\text{CD}, \text{club}))} \quad .$$

Let $n_{\text{children}}(\text{club})$ denote the number of children who are member of a specific club. Then the total number of children who play football in a specific CD is calculated as $n_{\text{football children}}(\text{CD}) = \sum_{\text{club}} n_{\text{children}}(\text{club}) \cdot P(\text{club} \rightarrow \text{CD})$.

## Estimating the club-size function

For every club, estimates of the total income, $e_{\text{income}}(\text{club})$, of every club, and the number of children, $n_{\text{children}}(\text{club})$ per club are available. We empirically validate that the functional relationship between the number of children per club and its total income can be reasonably approximated using parameters $\alpha > 0$ and $\beta > 0$

$$n_{\text{children}}(\text{club}) = \alpha \cdot \log(e_{\text{income}}(\text{club})) + \beta \quad .$$

We call this the *club-size function*.

**Socio-economic dependencies.** Sports club participation rates among children are influenced by socio-economic factors, and $\alpha$ may depend on the socio-economic indicators of the municipality. To determine which indicator is most influential with respect to the club-size function, $n_{\text{children}}(\text{club})$, we assign socio-economic indicators to each club based on the municipality it is located in. The list of triples—the income and number of children in the club, and the value of a specific socio-economic indicator of the club's surroundings—is written as $((e_{income_1}, n_{children_1}, ind_1), (e_{income_2}, n_{children_2}, ind_2), \dots, (e_{income_N}, n_{children_N}, ind_N))$, and $M_{ind}$ represents the median of the values in $(ind_1, \dots, ind_n)$. For each socio-economic property, the clubs are divided into two groups—those above and those below the median value $M_{ind}$. For these two groups of clubs, the following regression models are evaluated separately:

$$\forall i : ind_i \geq M_{ind} \quad n_{\text{children}_i} = \alpha_{\geq} \log(e_{\text{income}_i}) + \beta_{\geq} + \epsilon_i \tag{1}$$

$$\forall i : ind_i < M_{ind} \quad n_{\text{children}_i} = \alpha_{<} \log(e_{\text{income}_i}) + \beta_{<} + \epsilon_i \tag{2}$$

Here, $\alpha_{\geq}$ and $\alpha_{<}$ represent the regression coefficients, $\beta_{\geq}$ and $\beta_{<}$ are the intercepts, and $\epsilon_i$ is a random error variable. Differences between $\alpha_{\geq}$ and $\alpha_{<}$ are expressed as $Z$-scores, which measure the number of standard errors by which the measured effect size differs from zero. To derive a socio-economically stratified club-size function, we identify the indicator for which we empirically observe the largest differences in the correlations between club size and income across the different strata.

The number of children a sports club is expected to attract and add to its current members is calculated using the following formula:

$$n_{\text{add.children}}(\text{club})(e_{\text{add.funding}}) = \alpha(\text{club}) \cdot \log(e_{\text{curr. income}}(\text{club}) + e_{\text{add.funding}}(\text{club}))$$
$$-\log(e_{\text{curr. income}}) \quad .$$

Here, $\alpha(\text{club})$ represents the group-specific parameters, $e_{\text{current income}}$ denotes the current total income of the club, and $e_{\text{add.funding}}$ refers to the additional funding.

## Optimal funding strategy

An algorithm similar to gradient descent is employed to design a strategy that optimizes the allocation of a given amount of money, maximizing the expected number of additional children joining the club.

1. Calculate the current value of $n_{\text{add.children}}(\text{club})$ (1.000 EUR), for each club with its current available budget.

2. Each of the clubs with the maximum $n_{\text{add.children}}(\text{club})$ (1.000 EUR) receives a small amount of money, $\epsilon$ EUR.

3. Subtract $\epsilon$ EUR from the total available budget.

4. Go to step 1. Repeat until the entire budget is spent.

This algorithm helps us determine the funding scheme that distributes a fixed amount of funds across the clubs with the goal of maximizing the total number of additional children.

## Results

Fig 1(C) shows a map of the percentage of children playing football in every CD, according to the employed gravity model. It reveals an urban-rural differential. More children play football in urban areas than in rural ones, except in the city of Vienna and its surroundings.

## Socio-economically stratified club-size function

We empirically assess how socio-economic characteristics might impact the likelihood of clubs having more children relative to their income. In Table 1, we report the corresponding regression coefficients that encode how club size and income differ between clubs in regions with higher ($\alpha_{\geq}$) or lower ($\alpha_{<}$) indicator values.

The largest differences (Z-scores) are observed for the percentage of people who do not have Austrian citizenship, as well as the number of families per capita, the log-population density, and the average number of people per household. Some of these indicators are closely related; see Table 2, where we report pairwise Pearson correlation coefficients. From those four indicators, we choose the two that have the smallest correlation, suggesting that these indeed capture independent socio-economic determinants that influence one's likelihood to become a member of a sports club: the percentage of people who do not have Austrian citizenship and the average number of people per household. According to these socio-economic variables, the clubs are assigned to four groups based on whether the values of the socio-economic properties of the club are greater or less than their median. Table 3 shows a summary of the clubs in these four groups, along with information regarding the model parameter $\alpha$ in each group and the goodness-of-fit value $R^2$.

The larger the value of $\alpha$, the more additional children a club is assumed to attract for a given amount of funding. We find the smallest (largest) value of $\alpha$ for clubs in regions with above- (below-) median numbers of families and corporations. We find good agreement between the assumed club size function and the data, as indicated by an overall $R^2$ of 0.83.

**Table 1. For each socio-economic indicator, the results of the regressions for the club size function are shown.**
Two groups are considered, depending on whether the specific indicator for the municipality the club belongs to is greater than ($\geq$) or less than ($<$) the median for all clubs. The table displays the coefficients $\alpha_{\geq}$, $\alpha_{<}$, and their difference (Z-score). Indicators are ordered by the absolute value of the Z-score.

| Socioeconomic Property | Z | $\alpha_{\geq}$ | $\alpha_{<}$ |
|---|---|---|---|
| Percentage of people without Austrian citizenship | 5.79 | 55.04 | 19.60 |
| Number of families per capita | -5.74 | 21.38 | 57.19 |
| log(population density) | 5.63 | 54.01 | 21.73 |
| Average number of people per household | -4.79 | 28.96 | 58.24 |
| Percentage of commuters | -4.15 | 31.59 | 57.33 |
| Percentage, of the ages 15 to 64, employed | -3.95 | 35.69 | 59.81 |
| Percentage of the ages 15 and above, unemployed | 3.58 | 58.58 | 36.66 |
| Percentage of the ages 15 and above, with teritary education | 2.52 | 50.49 | 35.24 |
| Percentage of the ages 15 and above, with secondary education | -2.14 | 38.67 | 51.41 |
| Number of Corporations per capita | -1.92 | 42.23 | 53.97 |
| Percentage of people working at workplaces | 1.79 | 49.37 | 37.84 |
| Number of work places per capita | -1.59 | 44.42 | 54.21 |
| Percentage of ages 15 and above, with sec. or ter. education | -0.98 | 45.86 | 51.89 |

**Table 2. Pairwise Pearson correlation coefficients for the four socio-economic indicators that show the strongest effect on the number of children in a sports club.**

| | % without Austrian citizenship | #families per capita | log(pop. density) | #people per household |
|---|---|---|---|---|
| % without Austrian citizenship | 1.00 | -0.84 | 0.87 | -0.68 |
| # families per capita | -0.84 | 1.00 | -0.80 | 0.70 |
| log(pop.density) | 0.87 | -0.80 | 1.00 | -0.69 |
| # people per household | -0.68 | 0.70 | -0.69 | 1.00 |

**Table 3. For each tier of football clubs, we provide the estimated budget, the number of clubs, and the number of children (with its interquartile range, IQR) in each of the four groups, which are defined by having a high (above the median) or low (below the median) number of families and corporations per capita.** We also present the results of the regression analysis for the club-size function for each group. For $\alpha$ and $R^2$, see 2. The overall $R^2$ is 0.83.

| | | high % without AT cit. high #people per h.h. | | | high % without AT cit. low #people per h.h. | | | low % without AT cit. high #people per h.h. | | | low % without AT cit. low #people per h.h. | | |
|---|---|---|---|---|---|---|---|---|---|---|---|---|---|
| tier | budget [€] | clubs | children | IQR | clubs | children | IQR | clubs | children | IQR | clubs | children | IQR |
| 3 | 484125 | 15 | 162 | 144-198 | 23 | 171 | 105-246 | 2 | 69 | 39-99 | 3 | 152 | 112-192 |
| 4 | 110026 | 39 | 92 | 61-155 | 44 | 142 | 83-228 | 14 | 93 | 37-118 | 27 | 80 | 38-118 |
| 5 | 79065 | 65 | 89 | 50-115 | 60 | 114 | 75-174 | 35 | 53 | 39-74 | 71 | 39 | 12-76 |
| 6 | 75178 | 101 | 68 | 37-104 | 82 | 71 | 24-123 | 94 | 54 | 28-82 | 96 | 34 | 10-56 |
| 7 | 45286 | 88 | 61 | 32-81 | 90 | 39 | 0-70 | 133 | 50 | 25-70 | 106 | 42 | 14-78 |
| 8 | 48385 | 87 | 39 | 15-75 | 94 | 42 | 10-85 | 80 | 36 | 9-47 | 98 | 23 | 1-48 |
| 9 | 39952 | 44 | 54 | 31-71 | 23 | 70 | 44-104 | 60 | 60 | 41-73 | 23 | 32 | 0-70 |
| 10 | 34751 | 3 | 47 | 23-68 | 1 | 8 | 8-8 | 1 | 19 | 19-19 | 0 | 0 | 0 |
| $\alpha, R^2$ | | $\alpha = 44.9, R^2 = 0.94$ | | | $\alpha = 58.1, R^2 = 0.79$ | | | $\alpha = 15.0, R^2 = 0.34$ | | | $\alpha = 52.7, R^2 = 0.92$ | | |

In Fig 2, we demonstrate that the number of children per club scales logarithmically with the club's budget. The scaling (slope) depends on regional socio-economic indicators. Notably, we observe the steepest slope for clubs in regions with fewer corporations and families compared to those in other regions. Additionally, Fig 3 shows the number of children playing

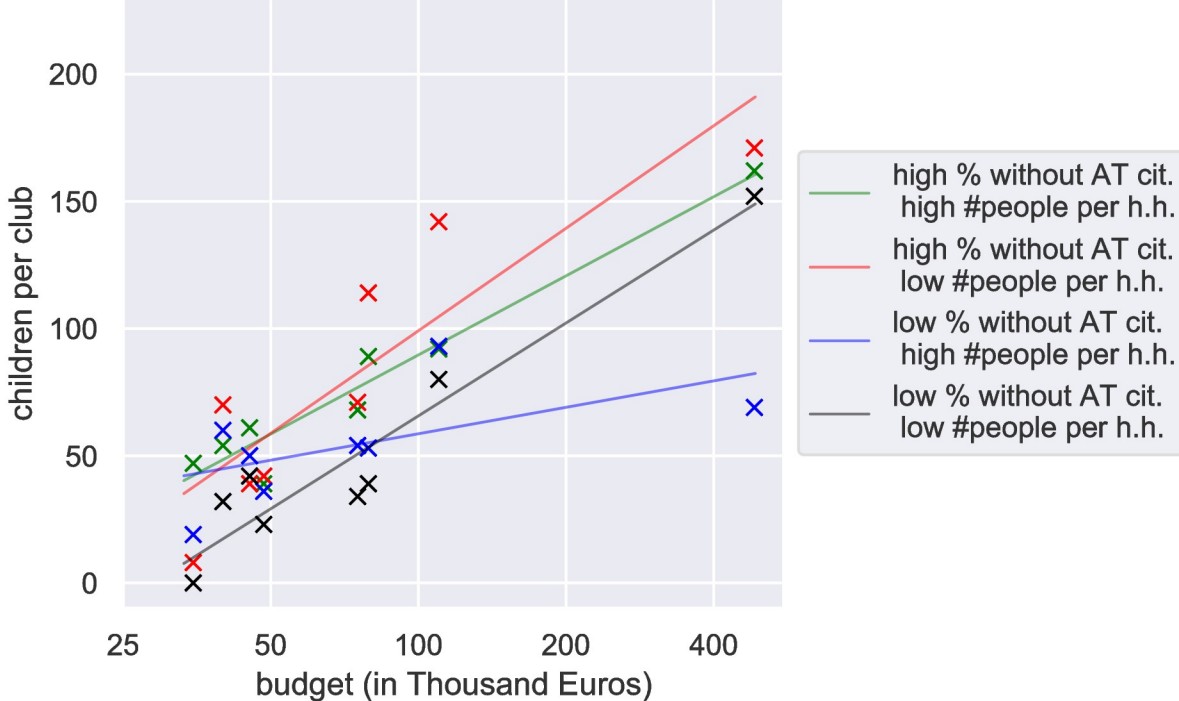

**Fig 2. The number of children per club increases logarithmically with its budget (the budget axis is scaled logarithmically).** Each cross marks the median number of children across all clubs at each competition level, for each of the four groups defined by a high or low percentage of people without Austrian citizenship, or by the average number of people per household in the club's region. The slope of the fit indicates the number of additional children a club in each group attracts per unit of additional funding. Notably, the slope is highest for clubs in regions with fewer corporations and families, meaning that a given amount of funding attracts more children in these regions than in any of the other groups.

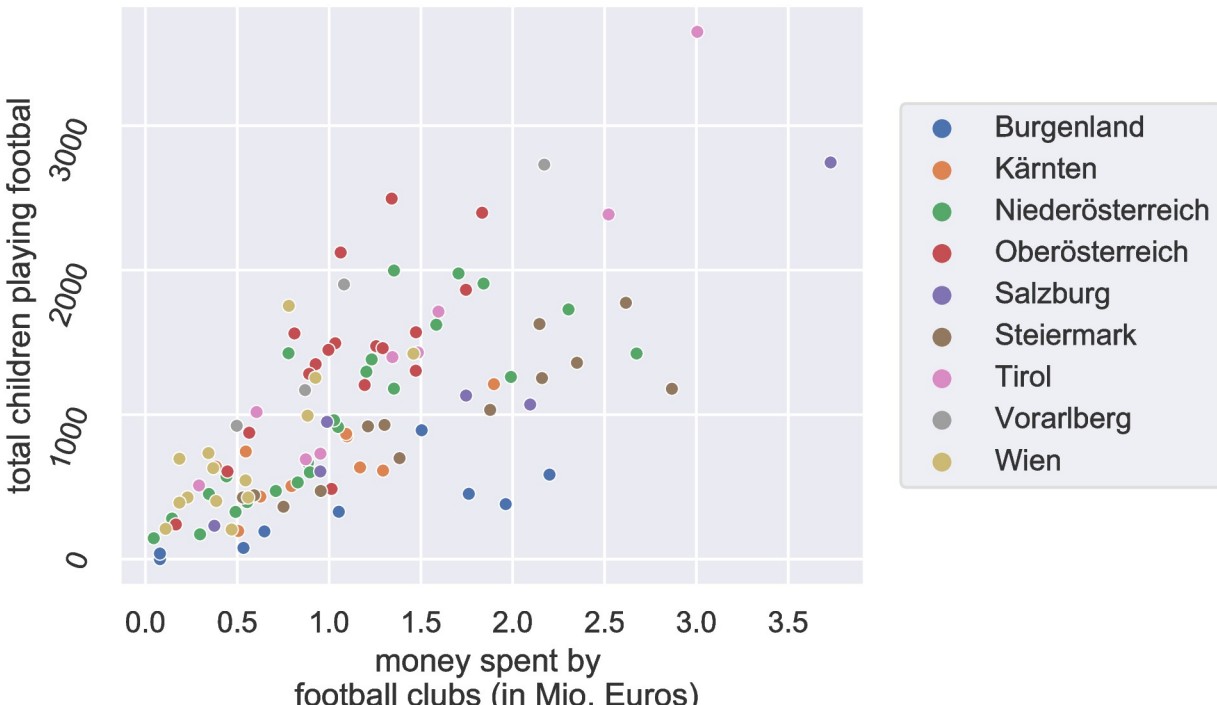

**Fig 3. The number of children who are members of football clubs per district increases with the total budget of all the football clubs in the district.** Note the significant variation in slopes across the different states of Austria. States handle funding policies autonomously.

football as a function of money spent by clubs in each district, where colors refer to federal states. A significant variation in slopes is observed across federal states.

The budget-activity curve allows us to benchmark different scenarios for the distribution of subsidies across sports clubs. Our central result presents the effectiveness of the four employed funding scenarios in Fig 4, which shows the number of additional children playing football as a function of subsidies spent on all the clubs. "In the first scenario, referred to as the *rich-get-richer* scenario (red), clubs receive additional subsidies as a percentage of their current income. In the second scenario, all clubs receive the same amount of money (the *all-equal* scenario, yellow). Because the club-size function is logarithmic, the all-equal scenario leads to more children playing football.

These two baseline funding strategies are compared with two optimized funding allocation schemes (green and yellow). In the optimized strategy, which considers social indicators (green), the number of expected additional children playing football per invested Euro is almost twice as large as for the *rich-get-richer* scenario and still substantially larger than in the *all-equal* case. A strategy that optimizes the distribution of funds based on club budgets but neglects socio-economic differences (blue) performs better than the all-equal case but substantially worse than the optimization that includes socio-economic factors. The scenarios shown in Fig 4 suggest that the larger the invested funds, the bigger the advantage of the optimization (bigger absolute difference in Fig 4). Regardless of the specific numbers, the simulations presented suggest that optimized funding strategies could offer significant advantages over traditional funding approaches.

We provide a dashboard that allows for user-specific designs of funding strategies dashboard (see Fig 5). In the dashboard, users can select sets of clubs and specify the amount of subsidies

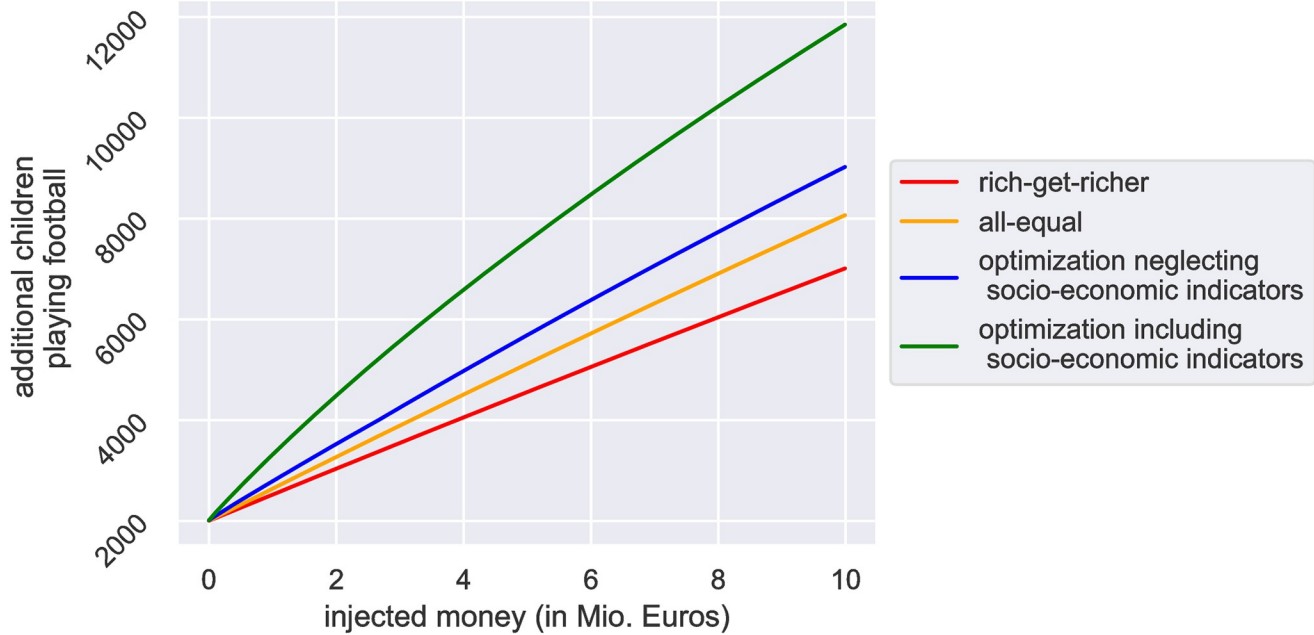

**Fig 4. Comparison of the outcome (additional children playing football) of the four considered funding schemes.** From best- to worst-performing, they are *optimization including socioeconomic indicators* (green), *optimization neglecting socioeconomic indicators* (blue), *all-equal*, and *rich-get-richer* (red).

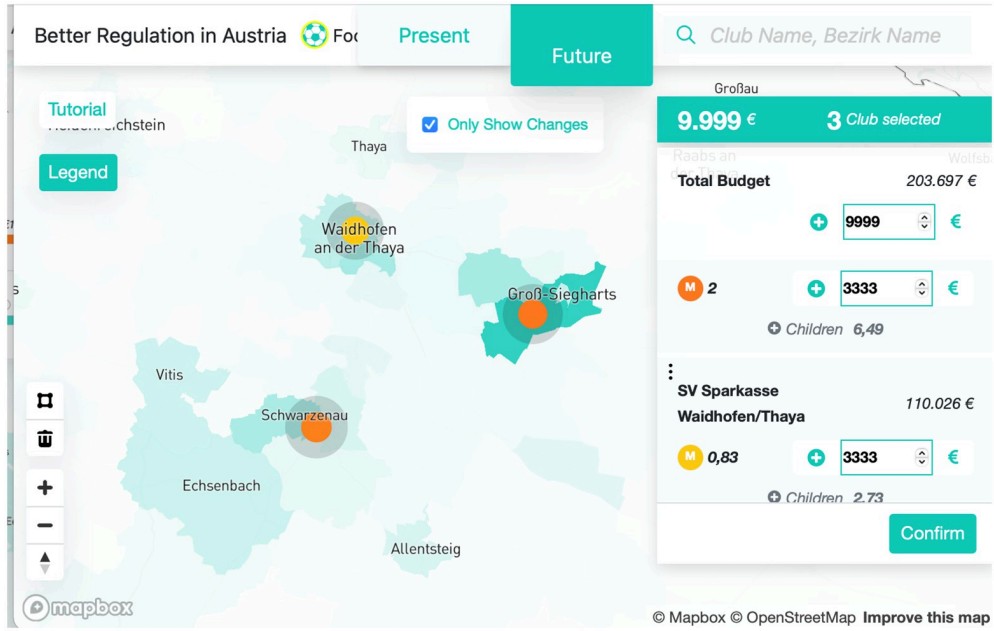

**Fig 5. This dashboard shows a model scenario where three clubs receive subsidies of 3,333 EUR each.** The football clubs are marked in yellow/orange (the color depends on the budget), and the turquoise shading of the CDs on the map indicates the expected number of additional children, along with the correct details about the clubs. On the right-hand panel, information about the clubs, such as total budget, additional subsidies, the *marginal activity rate* per EUR, and the number of additional expected children, is displayed.

to distribute among them. Based on the club-size function and the gravity model, the dashboard first calculates the number of additional children for each club and then for each CD.

## Conclusion

We propose a data-driven framework to identify optimal policy strategies for increasing physical activity in children by raising the number of sports club members. We assembled a country-wide dataset of 1,854 non-professional football clubs, including their locations, competition levels, budget estimates, and age data for their 225,000 members, including 111,000 children. Merging this dataset with publicly available data on the Austrian population and its socio-economic status enabled us to estimate how sports club funding enhances the attractiveness of a club to children. Compared to traditional methods, in which funds are distributed equally or proportionally to size, we find that optimized strategies significantly increase the number of children who can be attracted to join sports clubs.

The development of an optimized funding policy requires an estimate of how funding is linked to additional sports activity. We found a logarithmic budget–activity relation between the number of children and a club's budget, suggesting diminishing investement returns by funding larger clubs. The rate at which these returns diminish is influenced by the socio-economic and structural characteristics of the club's region, such as the number of families and corporations. For a total additional funding of 5 million Euros, the optimization neglecting socio-economic indicators attracts 34% fewer children than the optimization that includes socio-economic indicators. This result demonstrates the importance of tailoring funding strategies to specific socio-economic contexts. Socio-economic factors have been found to influence the effect of expenses on participation ratios [13]. Evidence shows that participation increases with household income and educational attainment [21–24]. Data from the Netherlands demonstrated a clear "dose-response" relationship between municipal sport expenditure and sport club participation rates, where lower income corresponds to a stronger increase in participation with additional funding [13]. Together, these findings suggest a 'compensation effect,' where government funding may offset the adverse impact of economic inequalities on physical activity. Our findings on the dependence of the budget-activity curve on socio-economic factors align with the existing literature.

In the present study, we do not have access to socio-economic indicators at an individual level. However, our high spatial resolution allowed us to adopt an ecological study design, relating regional differences in socio-economic factors to the relationship between available funds (proxied by the club budget) and participation rates. We found a particularly strong relationship between club budget and participation in areas with fewer corporations and workplaces, consistent with observed relationships between economic deprivation and the impact of funding in other countries. Our finding of a particularly strong gradient in the budget-participation relationship in regions with a high number of non-Austrian citizens is of particular concern, even though a strong correlation with the number of families per capita may have confounded this relationship. Further work is needed to understand how migration-related inequalities in access to sports facilities contribute to this strong gradient.

Our primary assumption in this work is that the empirical correlation captured in the budget-activity function reflects a causal relationship—at least partially. A potential driver of this relationship may be that better-funded clubs can attract children more easily compared to smaller clubs, even without additional external funding. The possibility that larger clubs with more children attract more funding is also plausible but is not considered here. Consequently, we implicitly assume that children's sports participation is supply-driven rather than demand-driven, meaning that if the supply of sports activities increases, there will always be sufficient

demand. More empirical work, ideally in the form of well-designed surveys, would be necessary to confirm these assumptions. In the absence of a model mechanism for saturation effects, our approach should not be applied to very high participation rates approaching 100%.

In terms of data, the study is mostly limited by missing information on the exact budgets of the individual clubs. As only competition-level estimates are available, this introduces a degree of uncertainty. Clubs often form associations to reach the number of players necessary for a team in a particular age group. Since it is impossible to reliably assign the players of these associations to specific clubs, this introduces uncertainties in estimating regional participation rates. However, we assume that these uncertainties do not introduce systematic biases in the estimation of club size functions.

Previous studies highlighted facility coverage as one of the driving factors of participation rates [9, 12]. Our data include implicit information on the facilities through the infrastructure costs in the clubs' budgets (e.g., football pitches, clubhouses). It remains to be seen in future work to what extent the presented results would change if club-size functions could be estimated while taking a club's available facilities into account.

We make this approach more accessible to decision-makers (and the public) through an interactive dashboard that allows for user-specific design of funding policies. However, tools like this are only one element among the requirements for evidence-based policymaking. In addition, qualified staff are needed to analyze data [25] and conduct plausibility checks. In the long run, in light of increasingly constrained budgets and public doubt about the efficacy of politics, it seems inevitable that evidence-based policymaking will be integrated into national performance management (outcome orientation). It can be regarded as international best practice and demonstrates the impact chain between long-term political goals and administrative measures at a technical level [26, 27].

Finally, proactive data management may require changes to existing laws that promote the development of central databases and interoperability [28]. Communication and cooperation between political institutions, public administration, and the scientific community can be enhanced by creating appropriate platforms for interdisciplinary exchange.

## Author Contributions

**Conceptualization:** Jan Hurt, Stefan Thurner, Peter Klimek.

**Data curation:** Jan Hurt.

**Formal analysis:** Jan Hurt.

**Funding acquisition:** Stefan Thurner.

**Investigation:** Jan Hurt, Nike Pulda, Peter Klimek.

**Project administration:** Thomas J. Lampoltshammer, Stefan Thurner, Peter Klimek.

**Supervision:** Stefan Thurner, Peter Klimek.

**Visualization:** Liuhuaying Yang, Johannes Sorger.

**Writing – original draft:** Jan Hurt, Peter Klimek.

**Writing – review & editing:** Jan Hurt, Thomas J. Lampoltshammer, Nike Pulda, Ursula Rosenbichler, Stefan Thurner, Peter Klimek.

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
