## [Decision Letter · Decision Letter 0]

31 Jul 2024

PONE-D-24-19945Evidence-based policy-making in sports funding using a data-driven optimization approach

PLOS ONE

Dear Dr. Hurt,

Thank you for submitting your manuscript to PLOS ONE. After careful consideration, we feel that it has merit but does not fully meet PLOS ONE’s publication criteria as it currently stands. Therefore, we invite you to submit a revised version of the manuscript that addresses the points raised during the review process.

Please note that we have only been able to secure a single reviewer to assess your manuscript. We are issuing a decision on your manuscript at this point to prevent further delays in the evaluation of your manuscript. Please be aware that the editor who handles your revised manuscript might find it necessary to invite additional reviewers to assess this work once the revised manuscript is submitted. However, we will aim to proceed on the basis of this single review if possible. 

The reviewer has assessed the manuscript and their comments are available below. They have made suggestions of some additional discussion points and also requested consistency between the introduction and the discussion. Please review their comments and make the appropriate revisions. 

We look forward to receiving your revised manuscript.

Kind regards,

Emma Campbell, Ph.D

Staff Editor

PLOS ONE

Journal Requirements:

"The project was funded by the European Union’s Structural Reform Support Programme

(SRSP) under Grant Agreement No. GA2020/023, and the Austrian Research Promotion

Agency FFG under P 882184."

"The authors received no specific funding for this work.”

5. We note that [Figures 1 and 5] in your submission contain [map/satellite] images which may be copyrighted. All PLOS content is published under the Creative Commons Attribution License (CC BY 4.0), which means that the manuscript, images, and Supporting Information files will be freely available online, and any third party is permitted to access, download, copy, distribute, and use these materials in any way, even commercially, with proper attribution. For these reasons, we cannot publish previously copyrighted maps or satellite images created using proprietary data, such as Google software (Google Maps, Street View, and Earth). For more information, see our copyright guidelines: http://journals.plos.org/plosone/s/licenses-and-copyright.

a. You may seek permission from the original copyright holder of Figures 1 and 5 to publish the content specifically under the CC BY 4.0 license.  

Reviewers' comments:

Reviewer's Responses to Questions

**Comments to the Author**

1. Is the manuscript technically sound, and do the data support the conclusions?

Reviewer #1: Yes

2. Has the statistical analysis been performed appropriately and rigorously? 

Reviewer #1: Yes

3. Have the authors made all data underlying the findings in their manuscript fully available?

Reviewer #1: Yes

4. Is the manuscript presented in an intelligible fashion and written in standard English?

Reviewer #1: No

5. Review Comments to the Author

Reviewer #1: The topic of the manuscript is important. For a long time the discussion related to the public financing of sports has been focused on the question: how to distribute the grants, so that they are distributed as effectively/efficiently as possible? The subject has been approached in this manuscript with versatile materials and analyses.

The whole is presented clearly and the results are consistent. In the following, however, I present some considerations that I hope the authors will take into account:

1. The topic is contextualized in the Introduction section to physical inactivity and obesity. The covid-19 pandemic is also mentioned. This frame is not discussed much in Discussion or elsewhere in the manuscript after the Introduction (or Abstract). At the very beginning, a hypothesis is even being developed between the pandemic, obesity and low physical activity. I would suggest that this framework will be removed and research should focused on how public sports funding should be targeted as a sport policy activity. The factors for that are the size of the clubs, funding, socio-economic status, etc. This is also related to the very important issue brought up in Discussion/Conclusion, i.e. knowledge based management and information/data as tool in political decision-making.

2. In the Conclusions/Discussion section, observations related to the material are highlighted. They are relevant. I suggest that, in the same context, authors also consider the possibilities of scaling this research to sports other than football, or possibly to the funding of non-organized sports that try to promote the movement of the population.

3. I also suggest that the grammar of the manuscript will be checked. The text is mostly at a good level, but sentence structures and general fluency can be improved.

I wish a good luck with the paper!

6. PLOS authors have the option to publish the peer review history of their article (what does this mean?). If published, this will include your full peer review and any attached files.

Reviewer #1: No

---

## [Author Response · Author response to Decision Letter 0]

13 Sep 2024

Dear Editor,

We sincerely appreciate the time and effort you and the reviewers invested in evaluating our manuscript. Their insightful feedback has been invaluable in refining our work. In response to the reviewers' comments, we have made substantial revisions to enhance the clarity and robustness of our manuscript. The key improvements include:

PLOS ONE Template and Style Requirements: We have formatted the manuscript according to the PLOS ONE LaTeX template and ensured that all elements follow the style guide provided in the links. This includes adherence to file naming conventions and other formatting standards.

Figures and Copyright: The maps in Figures 1 through 5 were generated using our own scripts and datasets. As such, these maps are under our own copyright, and no third-party data or copyrighted content has been used. Therefore, there are no copyright issues related to these figures, and they are fully compatible with the CC BY 4.0 license.

Funding Information: We have updated the financial disclosure statement to comply with the PLOS ONE guidelines. The updated statement includes the specific grant numbers, initials of the author who received each award, full names of the funding agencies, URLs to the sponsors’ websites, and a statement about the funders' role in the research. The revised financial disclosure statement is as follows:

Financial Disclosure Statement

The project was funded by the European Union's Structural Reform Support Programme (SRSP) under Grant Agreement No. GA2020/023 (received by S.T.; https://commission.europa.eu/about-european-commission/departments-and-executive-agencies/structural-reform-support/structural-reform-support-programme-srsp_en), and the Austrian Research Promotion Agency (FFG) under project number P 882184 (received by S.T.; https://www.ffg.at/en). The funders had no role in study design, data collection and analysis, decision to publish, or preparation of the manuscript.

Response to reviewer

Reviewer comment #1. "The topic is contextualized in the Introduction section to physical inactivity and obesity. The covid-19 pandemic is also mentioned. This frame is not discussed much in Discussion or elsewhere in the manuscript after the Introduction (or Abstract). At the very beginning, a hypothesis is even being developed between the pandemic, obesity and low physical activity. I would suggest that this framework will be removed and research should focus on how public sports funding should be targeted as a sport policy activity. The factors for that are the size of the clubs, funding, socio-economic status, etc. This is also related to the very important issue brought up in Discussion/Conclusion, i.e. knowledge-based management and information/data as tools in political decision-making."

We thank the reviewer for this valuable comment. We have carefully considered the suggestion and agree that the focus of the manuscript should be more clearly aligned with the targeting of sports funding as a policy activity. As such, we have revised the Introduction section to remove the references to COVID-19 and obesity. The updated Introduction now focuses on the key factors relevant to sports funding policy, such as the size of sports clubs, funding allocations, and socio-economic status.

Reviewer comment #2. "In the Conclusions/Discussion section, observations related to the material are highlighted. They are relevant. I suggest that, in the same context, authors also consider the possibilities of scaling this research to sports other than football, or possibly to the funding of non-organized sports that try to promote the movement of the population."

We thank the reviewer for this suggestion. We considered expanding the research to include other sports; however, due to the unavailability of data on the number of children participating in sports clubs besides football, this was not feasible at this time. Our current dataset includes detailed information only for football clubs. Without similar data for other sports, it would be difficult to apply the same methodological framework. Nevertheless, we acknowledge that scaling the research to other sports or even non-organized sports activities would be a valuable future direction."

Reviewer comment #3. "I also suggest that the grammar of the manuscript will be checked. The text is mostly at a good level, but sentence structures and general fluency can be improved."

We thank the reviewer for this suggestion. We have carefully proofread the manuscript and made several revisions to improve fluency and readability.

---

## [Decision Letter · Decision Letter 1]

2 Oct 2024

Evidence-based policy-making in sports funding using a data-driven optimization approach

PONE-D-24-19945R1

Dear Dr. Hurt,

We’re pleased to inform you that your manuscript has been judged scientifically suitable for publication and will be formally accepted for publication once it meets all outstanding technical requirements.

Kind regards,

Gábor Vattay, PhD, DSc

Academic Editor

PLOS ONE

Additional Editor Comments (optional):

The study presents original research on optimizing sports funding using a data-driven approach, which has not been published elsewhere. The data analyses are conducted rigorously, with clear methodologies provided, including the use of a gravity model and optimization algorithms. The conclusions are appropriately drawn and well-supported by the presented data, demonstrating a clear impact of funding strategies on children's participation in sports.

The manuscript is written in clear, standard English, making it easy to understand for readers.

The study adheres to ethical standards and provides sufficient data availability through public dashboards.

The authors revised the introduction focusing the manuscript on the core issue of public sports funding as a policy activity. This shift highlights the factors such as the size of sports clubs, funding allocations, and socio-economic status, aligning better with the policy-oriented focus that the reviewer recommended. They acknowledged that scaling the research to other sports or non-organized sports would be valuable. However, due to a lack of data for other sports, such expansion was not feasible at this stage. They also conducted a careful proofread, making several revisions to improve the overall fluency and readability of the manuscript, ensuring clarity.

Reviewers' comments:

Reviewer's Responses to Questions

**Comments to the Author**

1. If the authors have adequately addressed your comments raised in a previous round of review and you feel that this manuscript is now acceptable for publication, you may indicate that here to bypass the “Comments to the Author” section, enter your conflict of interest statement in the “Confidential to Editor” section, and submit your "Accept" recommendation.

Reviewer #1: All comments have been addressed

2. Is the manuscript technically sound, and do the data support the conclusions?

Reviewer #1: Yes

3. Has the statistical analysis been performed appropriately and rigorously? 

Reviewer #1: Yes

4. Have the authors made all data underlying the findings in their manuscript fully available?

Reviewer #1: Yes

5. Is the manuscript presented in an intelligible fashion and written in standard English?

Reviewer #1: Yes

6. Review Comments to the Author

Reviewer #1: I appreciate the opportunity to reevaluate the manuscript. The authors have taken into account the comments I made in the original manuscript.

7. PLOS authors have the option to publish the peer review history of their article (what does this mean?). If published, this will include your full peer review and any attached files.

Reviewer #1: No

---

## [Editor Report · Acceptance letter]

18 Oct 2024

PONE-D-24-19945R1 

PLOS ONE

Dear Dr. Hurt, 

I'm pleased to inform you that your manuscript has been deemed suitable for publication in PLOS ONE. Congratulations! Your manuscript is now being handed over to our production team.

Kind regards, 

on behalf of

Dr. Gábor Vattay 

Academic Editor

PLOS ONE